# Convergent eusocial evolution is based on a shared reproductive groundplan plus lineage-specific plastic genes

Michael R. Warner[1], Lijun Qiu[2], Michael J. Holmes[2,3], Alexander S. Mikheyev [2,4] & Timothy A. Linksvayer [1]

Eusociality has convergently evolved multiple times, but the genomic basis of caste-based division of labor and degree to which independent origins of eusociality have utilized common genes remain largely unknown. Here we characterize caste-specific transcriptomic profiles across development and adult body segments from pharaoh ants (*Monomorium pharaonis*) and honey bees (*Apis mellifera*), representing two independent origins of eusociality. We identify a substantial shared core of genes upregulated in the abdomens of queen ants and honey bees that also tends to be upregulated in mated female flies, suggesting that these genes are part of a conserved insect reproductive groundplan. Outside of this shared groundplan, few genes are differentially expressed in common. Instead, the majority of the thousands of caste-associated genes are plastically expressed, rapidly evolving, and relatively evolutionarily young. These results emphasize that the recruitment of both highly conserved and lineage-specific genes underlie the convergent evolution of novel traits such as eusociality.

[1] University of Pennsylvania, Philadelphia, PA 19104, USA. [2] Okinawa Institute of Science and Technology, Okinawa 904-0495, Japan. [3] School of Life and Environmental Science, University of Sydney, Sydney 2006, Australia. [4] Research School of Biology, Australian National University, Canberra 0200, Australia. Correspondence and requests for materials should be addressed to M.R.W. (email: michael.ryan.warner@gmail.com)

The degree to which convergent phenotypic evolution involves the same sets of genes or pathways is a major unanswered question[1]. Comparative genomic studies indicate that parallel adaptive changes in the protein-coding sequences of the same genes are frequently associated with the evolution of convergent phenotypes in closely related populations and species[2,3]. Decades of research in evolutionary developmental biology also emphasize that changes in the expression of a relatively small toolkit of deeply conserved genes are often associated with convergently evolved phenotypes in distantly related species[4]. Alternatively, convergent phenotypic evolution between lineages could involve distinct subsets of genes in each lineage, including taxonomically restricted genes, genes which have no detectable orthology outside of a given lineage[5]. Taxonomically restricted genes have been shown to be important for lineage-specific evolutionary novelties[6], but their relative contribution to the evolution of convergent phenotypes is unknown.

The evolution of eusociality in several insect lineages (e.g., ants, honey bees, vespid wasps, and termites) provides a striking example of convergent phenotypic innovation[7]. Eusocial insect societies are founded upon a novel caste polyphenism, in which reproductive queen and non-reproductive worker female castes develop from the same genome, depending mainly on socially regulated nutritional inputs[8,9]. Within the worker caste, further specialization often occurs as individuals age and progress through a series of tasks, including nursing and foraging[7].

Polyphenic traits are often thought to evolve from pre-existing developmental plasticity[10]. Leading hypotheses for the evolution of caste-based division of labor in social insects also stress the use and modification of highly conserved developmental and physiological mechanisms[11–15]. For example, the reproductive and non-reproductive phases of ancestral solitary insects are thought to have been decoupled to produce reproductive and non-reproductive castes[11,16], and worker division of labor is similarly thought to be derived from the decoupling of the ancestral reproductive cycle[13,16,17]. Along the same lines, it has been suggested that the convergent evolution of novel social behavior involves changes to the regulation of a core toolkit of genes underlying highly conserved physiological processes, such as metabolism[14,15].

Studies focused on candidate genes underlying the genetic basis of caste-based division of labor within individual eusocial species have often found support for the importance of highly conserved genes and pathways associated with reproduction and metabolism. For example, worker division of labor in honey bees is regulated by interactions between juvenile hormone, vitellogenin, and insulin/TOR signaling pathways[13,17,18]. Similar pathways also play key roles in regulating division of labor between queen and worker castes in both ants and honey bees, though the mechanistic details vary[19–22]. While comparative genomic and transcriptomic studies have often similarly emphasized common general functions, such as metabolism, such studies have thus far only identified very small sets of specific genes associated with the convergent evolution of caste, worker behavior, or eusociality in independent lineages[15,23–27]. Alternatively, many transcriptomic studies have argued for the importance of taxonomically restricted genes for the evolution of caste-based division of labor[28–34]. It is unclear if the lack of common specific genes is due to biological differences between the species or methodological details, because studies in each species were not designed, conducted, or analyzed in parallel.

Previous work has mainly focused on identifying whether there is significant overlap of genes or gene pathways associated with caste-based division of labor between independent lineages[23,26,27], but there has been little effort to quantify the relative importance of shared versus unshared genes to the convergent evolution of caste-based division of labor. Most of these studies have either focused on the brain or whole-body samples[15,22,24,26,33,35–37], although expression bias between queens and workers has been shown to be dependent upon developmental stage and tissue type[34,38–40]. Finally, the transcriptomic signatures of reproductive physiology are strongest in the abdomen[34,41], the location of reproductive organs, but no past study has explicitly compared caste bias in abdominal tissues in species from lineages representing independent origins of eusociality.

Here, we present a comprehensive developmental transcriptomic data set investigating gene expression associated with reproductive caste and age-based worker division of labor in the pharaoh ant (*Monomorium pharaonis*) and the honey bee (*Apis mellifera*). We focus on these two study species because they represent two independent origins of eusociality in the ant and corbiculate bee lineages[42] as well as two independent elaborations of eusociality, each characterized by strong queen-worker dimorphism and age-based worker division of labor[33,43]. We perform all sampling, sequencing, and analysis for the two species in parallel to maximize compatibility between the data sets. We leverage this extensive data set to quantify in an unbiased manner the relative contribution of differential expression of shared versus distinct genes at each life stage and tissue to the convergent evolution of caste-based division of labor. We identify a large group of genes which are associated with queen abdomens in both eusocial species and tend to be female biased in *Drosophila melanogaster*. Outside of this shared core, few genes are differentially expressed in both species in the same tissue or developmental stage, and genes with high degrees of caste-biased expression tend to be weakly constrained in terms of expression profile and sequence evolution.

## Results

**Study design.** We constructed two large, parallel transcriptomic data sets in honey bees and pharaoh ants spanning caste development as well as adult tissues separated by reproductive caste (queens versus workers), behavior (nurse workers versus forager workers), and sex (queens and workers versus males). In total, we constructed 177 mRNA-sequencing libraries across 28 distinct sample types for each species (Supplementary Table 1).

**Differential expression between queens and workers.** To identify genes associated with caste development and adult caste dimorphism, we performed differential expression analysis between queens and workers at each developmental stage and adult tissue, separately for each species. The number of differentially expressed genes (DEGs) between queens and workers increased throughout development, peaking in the adult abdomen (Fig. 1a). In all tissues and stages, the majority of caste-associated DEGs in one species were either not differentially expressed or did not have an ortholog in the other species (Fig. 1a; Supplementary Fig. 1a; Supplementary Table 2). The magnitude of gene-wise caste bias (as measured by $\log_2$ fold change between queen and worker samples) was weakly positively correlated between ant and honey bee orthologs in all three adult tissues, with the strongest correlation in the abdomen, but uncorrelated or negatively correlated in all larval and pupal stages (Supplementary Fig. 2; Pearson correlation; $r_{head} = 0.089$; $r_{thorax} = 0.161$; $r_{abdomen} = 0.275$; $N = 7460$ 1:1 orthologs; $P < 0.001$ in all cases). The top enriched Gene Ontology (GO) terms for caste-associated DEGs in each species were dominated by metabolism, signaling, and developmental processes (Supplementary Tables 3, 4).

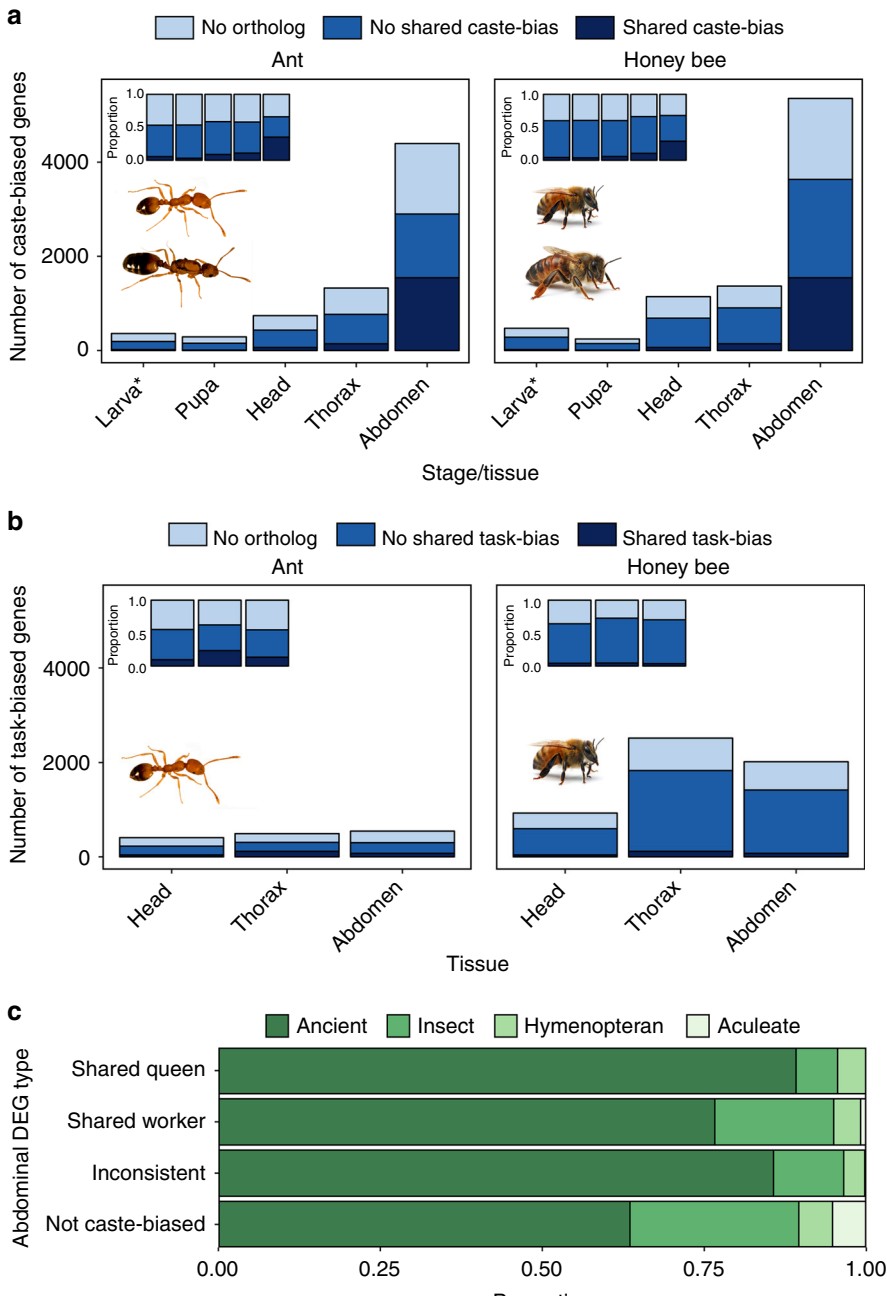

**Fig. 1** Patterns of caste-biased expression in pharaoh ants and honey bees. The number of differentially expressed genes (FDR < 0.1) between (**a**) queens and workers and (**b**) nurses and foragers at each developmental stage or tissue in ants (left) and honey bees (right). "Head", "thorax", and "abdomen" refer to body segments of adults, while "pupa" and "larva" refer to whole bodies. "No ortholog" refers to genes for which no 1:1 ortholog exists (either due to apparent duplication or complete lack or orthology), "not shared caste/task bias" refers to genes for which 1:1 orthologs can be identified but are only differentially expressed in one species, and "shared caste/task" bias refers to genes for which 1:1 orthologs are differentially expressed in both species. Insets show the proportion of each category of gene out of all differentially expressed genes at that stage or tissue. **c** Proportion of abdominal DEGs by estimated evolutionary age (shading). "Shared queen/worker" indicates genes upregulated in queen or workers of both species. *: the category "larva" represents differential expression across larvae of all stages for which caste can be identified (second to fifth larval stage). Source data are provided as a Source Data file. Photos were taken by Luigi Pontieri (pharaoh ants) and Alex Wild (honey bees)

**Differential expression between nurses and foragers**. Both honey bees[43] and pharaoh ants[33] exhibit age-based worker division of labor, in which younger individuals tend to specialize on nursing and other within-nest activities and older individuals specialize on foraging. To identify genes associated with age-based worker division of labor, we performed differential expression analysis between nurses and foragers in each adult tissue,

separately for each species. In general, there were very few behavioral DEGs shared between the two species (Fig. 1b; Supplementary Fig. 1b; Supplementary Table 5). Gene-wise $\log_2$ fold change between nurses and foragers was significantly but weakly correlated across ant and honey bee orthologs (Supplementary Fig. 3; Pearson correlation; $r_{head} = 0.070$, $P_{head} < 0.001$; $r_{thorax} = 0.031$, $P_{thorax} = 0.008$; $r_{abdomen} = 0.051$, $P_{abdomen} < 0.001$; $N = 7460$

1:1 orthologs). The top enriched GO terms for behavioral DEGs in each species were dominated by metabolism and developmental processes (Supplementary Tables 6, 7).

**Shared abdominal caste bias in ancient genes**. For the most part, our results indicate distinct genes are associated with caste and worker division of labor in honey bees and ants. However, approximately one-third of the abdominal caste-associated DEGs were common to both species (Fig. 1a; 1545 shared DEGs, comprising 35% [1545/4395] of ant DEGs, and 29% [1545/5352] of honey bee DEGs). Most shared abdominal differential expression was the result of shared queen bias: 56% (858/1545 genes) of shared abdominal caste-associated DEGs were upregulated in queen abdomens in both species, compared with 22% (338/1545) that were worker-upregulated in both species and 23% (349/1545) that reversed direction (i.e., were queen biased in one species and worker biased in the other). Shared abdominal caste-associated DEGs were more likely to be identified as evolutionarily ancient in comparison with non-biased genes (Fig. 1c; Supplementary Fig. 1c; Fisher Test; F = 3.41, $P < 0.001$). Furthermore, abdominal DEGs with shared queen bias were more likely to be identified as ancient than DEGs with shared worker bias (Fig. 1c; Fisher Test; F = 2.51, $P < 0.001$). In general, the evolutionary age of genes was associated with expression bias between castes, though the direction of the effect was not consistent across all tissues and stages (Supplementary Fig. 4).

We next tried to put the seemingly large proportion of shared abdominal caste-associated DEGs (35% for ants and 29% for honey bees) into context. We compared the proportion of genes that were differentially expressed across embryonic and larval development in both species, given that the molecular mechanisms of development are thought to be highly conserved[44]. We identified 6089 and 6225 developmental DEGs in ants and honey bees, respectively, including 2544 shared DEGs, representing 42% (2544/6089) and 41% (2544/6255) of the total developmental DEGs in each species (Supplementary Fig. 5).

To identify which of the thousands of abdominal DEGs found in each species are particularly important for queen abdominal expression (and presumably function), we performed gene co-expression analysis, separately for each species. We focused on modules specifically associated with queens because the majority of shared DEGs were queen upregulated. We identified a module of genes, specifically associated with queen abdominal expression in each species ($N = 1006$ genes in the module for ants, $N = 1174$ genes for honey bees). We identified hub genes in each module ($N = 92$ genes in ants, $N = 94$ genes in honey bees), genes which are centrally connected in networks and strongly associated with queen abdominal expression[45]. Many annotated hub genes are inferred to have functions associated with reproduction and maternal effects (Supplementary Tables 8, 9), including genes with known roles in caste determination, such as vitellogenin (*Vg* receptor was identified in each species)[20] and *vasa*[46], while others are important maternal proteins, such as *Smaug*[47] and *ovo*[48]. Furthermore, genes for which *Drosophila melanogaster* orthologs are known to function in oogenesis (based on FlyBase Gene Ontology[49]) were more highly connected within the queen abdominal modules than genes not associated with oogenesis (Supplementary Fig. 6) for honey bees (Wilcoxon test; $N = 649$; $P < 0.001$), though not for ants ($N = 542$; $P = 0.114$). Finally, we identified 181 genes which were present in the queen abdominal module of both species. These genes tended to be queen biased (78.5% [142/181] upregulated in queens of both species) and were more centrally located within modules than genes found in only one species-specific module (Fig. 2c, d).

**Caste bias is in part derived from ancestral sex bias**. Given that our co-expression analysis indicated that many important queen-upregulated genes are associated with oogenesis and overall female reproduction, we reasoned that caste-biased expression would be linked to sex-biased expression (i.e., expression differences between reproductive females and males). Indeed, there was a positive correlation between gene-wise log₂ fold change between queen and worker abdomens and gene-wise log₂ fold change between queen and male abdomens in both honey bees and pharaoh ants (Fig. 3 a, b). In addition, sex bias itself was correlated between species (Fig. 3c). The correlation of caste bias and sex bias was not restricted to the abdomen, as there were similar highly significant effects when comparing transcriptomic profiles in the head and thoracic tissues, albeit with weaker effect sizes (Supplementary Fig. 7).

Given the association between shared caste bias and sex bias within pharaoh ants and honey bees, we hypothesized that these shared caste-biased genes were derived from conserved pathways that also underlie sexual dimorphism for reproductive physiology in distant relatives. To test this hypothesis, we estimated sex-biased expression of orthologs in the fruit fly *D. melanogaster* using available data from male and mated female whole bodies[44]. Shared queen-biased abdominal DEGs tended to be upregulated in females in *D. melanogaster* (Fig. 3d; one-sided binomial test for likelihood of shared queen-biased DEGs having log₂ fold change > 0; $P < 0.001$; $N = 566$ shared queen DEGs), while shared worker-biased abdominal DEGs tended to be upregulated in males (binomial test; $P < 0.001$; $N = 160$ shared worker DEGs), indicative of shared queen (social insects) and female (fly) downregulation. Though we detected few shared caste-associated DEGs in the head and thorax ($N = 38$ and $N = 64$, respectively), these DEGs showed the same pattern, where orthologs of queen-biased DEGs were significantly more female biased in *D. melanogaster* than orthologs of worker-biased DEGs (Supplementary Fig. 8).

**Expression plasticity is correlated between species**. While we have emphasized the conservation of abdominal differential expression between queens and workers in pharaoh ants and honey bees, differential expression based on either reproductive caste or worker division of labor was largely not shared between species (Fig. 1). Furthermore, genes were often differentially expressed across many stages and tissues, sometimes in opposite directions (Supplementary Fig. 9; e.g., upregulated in queen heads, but downregulated in queen abdomens). To quantify the degree to which genes exhibited biased expression according to reproductive caste across all developmental stages and tissues, we calculated gene-wise overall caste bias in each species, where we defined overall caste bias as the Euclidean distance of log₂ fold change across all queen/worker comparisons[49]. Similarly, we defined overall behavior bias as the Euclidean distance of log₂ fold change across all nurse/forager comparisons, separately for each species.

Across 1:1 orthologs, overall caste bias measured in ants was correlated to overall caste bias measured in honey bees (Supplementary Fig. 10a; Spearman correlation; rho = 0.454, $P < 0.001$), and overall behavior bias was similarly correlated between species (Supplementary Fig. 10b; Spearman correlation; rho = 0.221, $P < 0.001$). Within species, overall caste and behavior bias were also correlated to each other (Supplementary Fig. 11; Spearman correlation; ants: rho = 0.549, $P < 0.001$; honey bees: rho = 0.642, $P < 0.001$). This indicates that plasticity in gene expression is correlated across contexts (caste versus behavior) and species. GO terms associated with high overall caste bias were largely linked to metabolism, while those associated with high

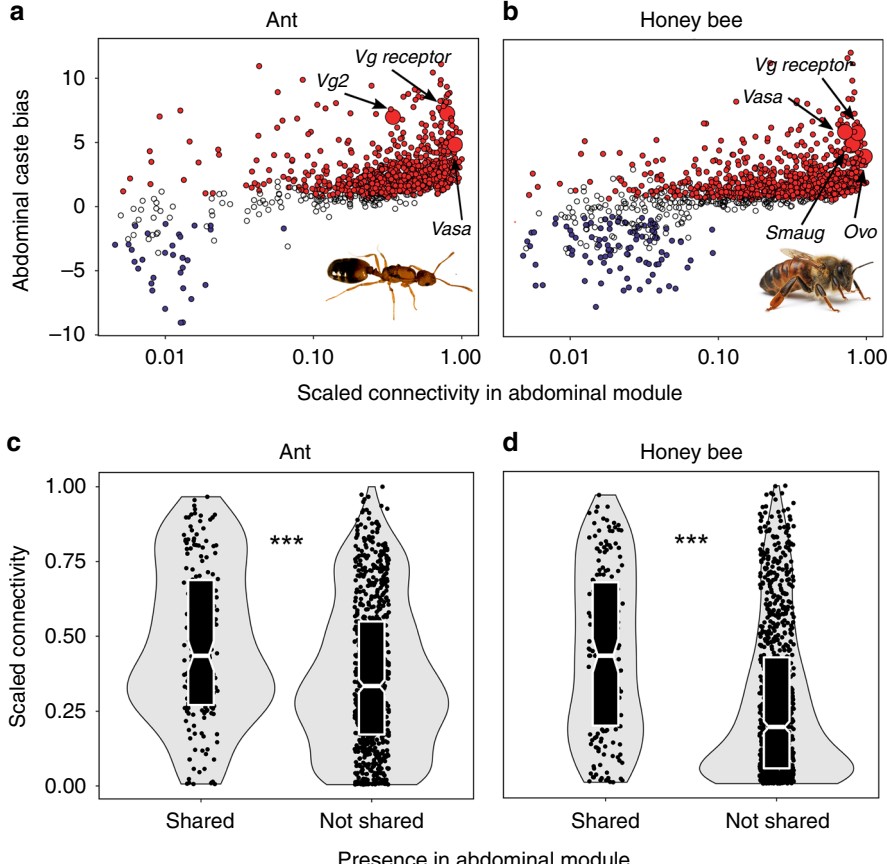

**Fig. 2** Genes with shared queen-biased expression are core network elements related to reproduction. Abdominal caste bias ($\log_2$ fold-change queen versus worker) is correlated with connectivity within the queen-abdomen module in (**a**) ants (Spearman correlation; rho = 0.536, $P < 0.001$) and (**b**) honey bees (Spearman correlation; rho = 0.617, $P < 0.001$). Genes upregulated in queens are in red, while genes upregulated in workers are in blue. Connectivity is proportional to the most highly connected gene in the module. Connectivity within the queen abdominal module is higher for genes found in the module for both species (shared) versus genes found in the module for only one species (not shared) in (**c**) ants and (**d**) honey bees. Middle line represents median values, outer edges of boxplot represent upper and lower quartiles, and whiskers represent a deviation of 1.5*(interquartile range) from the upper and lower quartiles. Source data are provided as a Source Data file. ***$P < 0.001$ (Wilcoxon test). Photos were taken by Luigi Pontieri (pharaoh ant) and Alex Wild (honey bee)

overall behavior bias were largely linked to developmental processes (Supplementary Table 10).

**Characteristics of genes associated with caste and behavior**. We compared overall caste bias and overall behavior bias to gene age, evolutionary rate, network connectivity, and tissue specificity to understand the general features of genes commonly associated with caste (queen versus worker) or behavior (nursing versus foraging). Genes with younger estimated evolutionary ages tended to exhibit higher overall caste bias (Fig. 4a, b) and behavior bias (Supplementary Fig. 12a, b) compared in particular to ancient genes (gamma GLM; ant caste bias: $\chi^2 = 900.19$, honey bee caste bias: $\chi^2 = 1412.80$, ant behavior bias: $\chi^2 = 316.36$, honey bee behavior bias: $\chi^2 = 877.43$; $P < 0.001$ for all cases; $N = 10520$ in ant, $N = 10011$ in honey bees). Genes that were loosely connected (representing peripheral network elements) in co-expression networks constructed across all samples tended to exhibit more caste and behavior bias in comparison with highly connected genes (Fig. 4c, d; Supplementary Fig. 12c, d). Similarly, genes with high tissue specificity across 12 honey bee tissues tended to exhibit higher values of caste and behavior bias in honey bees compared with more pleiotropic, ubiquitously expressed genes (Supplementary Fig. 13), where tissue specificity was calculated using available data[32]. Finally, genes that were

rapidly evolving (as estimated by dN/dS) tended to exhibit higher levels of caste and behavior bias (Fig. 4e, f; Supplementary Fig. 12e, f). Importantly, while expression is correlated to overall caste and behavior bias, these results remain highly significant when expression level is controlled for in partial correlation analyses (Supplementary Table 11).

## Discussion

Caste-based division of labor within social insect colonies is hypothesized to be derived from conserved pathways regulating reproduction[11,13,16,17]. In this study, we identified a large set (~1500) of genes with shared caste-biased abdominal expression in pharaoh ants and honey bees (Fig. 1a), including many annotated genes with known roles in reproduction, such as the *vitellogenin* receptor[20] and *ovo*[48]. Our results are consistent with the notion that caste-biased genes are derived from ancient plastically expressed genes underlying female reproduction, as genes upregulated in queen abdomens of both ants and honey bees tended to also be female biased in the distant insect relative *Drosophila melanogaster* (Fig. 3d). Previous studies had failed to find large sets of genes repeatedly used for eusocial evolution[23–27], but no previous comparative study investigated caste-biased expression in the abdomen.

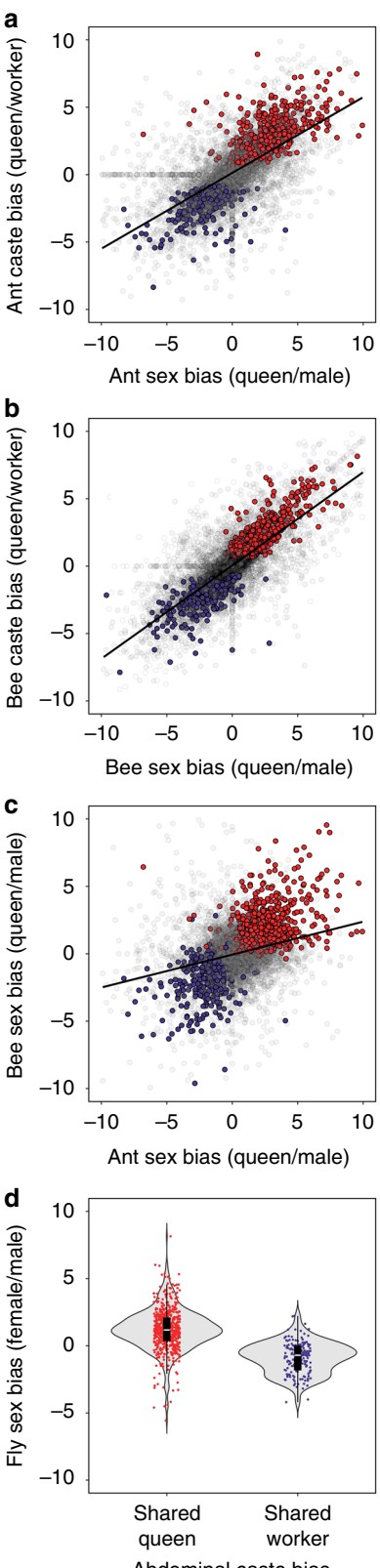

**Fig. 3** Caste bias is linked to sex bias. Abdominal caste bias (queen vs. worker log$_2$ fold change) is correlated to abdominal sex bias (queen vs male log$_2$ fold change) in **a**) *M. pharaonis* (Spearman correlation; rho = 0.715, $P <$ 0.001) and **b**) *A. mellifera* (Spearman correlation; rho = 0.774, $P <$ 0.001) and abdominal sex bias is correlated between the two species (Spearman correlation; rho = 0.280, $P <$ 0.001) (**c**). Red indicates shared queen-biased abdominal DEGs, while blue indicates shared worker-biased abdominal DEGs. Gray indicates genes that did not exhibit shared expression patterns or were not differentially expressed. Lines in **a**–**c** indicate the trendline of a linear model. **d** Shared queen-biased abdominal DEGs tend to be female biased in *D. melanogaster*, while shared worker-biased abdominal DEGs tend to be male biased in *D. melanogaster* (likely reflecting downregulation in females). Middle line represents median values, outer edges of boxplot represent upper and lower quartiles, and whiskers represent a deviation of 1.5*(interquartile range) from the upper and lower quartiles. Source data are provided as a Source Data file

shared ancestry and the deep conservation of developmental mechanisms[44,51]. The similar level of overlap for caste-associated genes points to the large-scale recruitment of pre-existing developmental and physiological machinery during the independent evolution of caste-based division of labor in ant and honey bee lineages. In addition, the association between sex bias in *D. melanogaster* and shared caste bias in social insects extended to the head and thorax (Supplementary Fig. 8), and caste- and sex bias were correlated within species (Supplementary Fig. 7). This indicates that although the strongest signature of overlap in caste-biased expression occurred in the abdomen, the association of caste- and sex bias is not simply driven by the presence of ovaries but rather due to shared female reproductive physiology that is largely conserved across insects.

While reproductive caste in complex eusocial societies such as ants and honey bees is typically fixed in adulthood, the tasks performed by workers (specifically, nursing versus foraging) change over the course of the worker's adult lifetime[18,33]. This plastic behavioral change is known to be accompanied by a wide range of physiological changes and is regulated at least in part by conserved physiological pathways, for example, those involving insulin signaling, juvenile hormone, and vitellogenin[18,21]. However, we identified few genes that were commonly differentially expressed between nurses and foragers in honey bees and pharaoh ants (Fig. 1b), and the proportion of shared genes was much lower in comparison with genes underlying abdominal differences between queens and workers. This could reflect a combination of different roles of nurses and foragers between lineages[7] as well as differences in the precise molecular relationships between these conserved pathways[20,52,53]. Nonetheless, we did identify a number of enriched Gene Ontology categories associated with development and metabolism in each species (Supplementary Tables 6, 7), which is consistent with the notion that the transition from nurse to forager is essentially a developmental process, and that common molecular pathways may provide the raw genetic material for social evolution[14,26,27].

Conserved factors or pathways clearly play important roles in aspects of caste development and function as well as the transition from nursing to foraging, but our results and other studies indicate that the majority of the full transcriptomic architecture associated with caste and age polyethism is not shared between species[26,29,33,36,37]. This lineage-specific architecture comprises large groups of both orthologous genes with different expression patterns and taxonomically restricted genes (Fig. 1a, b). In contrast to the low amount of context-specific overlap in differential expression, the overall degree of caste-associated plastic expression across stages and tissues (overall caste bias) was correlated

The large overlap for abdominal caste-associated genes is notable because honey bees and ants last shared a common ancestor ~160 million years ago[50], and this overlap is nearly as much as we see for genes that were differentially expressed across developmental stages (Supplementary Fig. 5). Shared developmental molecular mechanisms are presumably simply due to

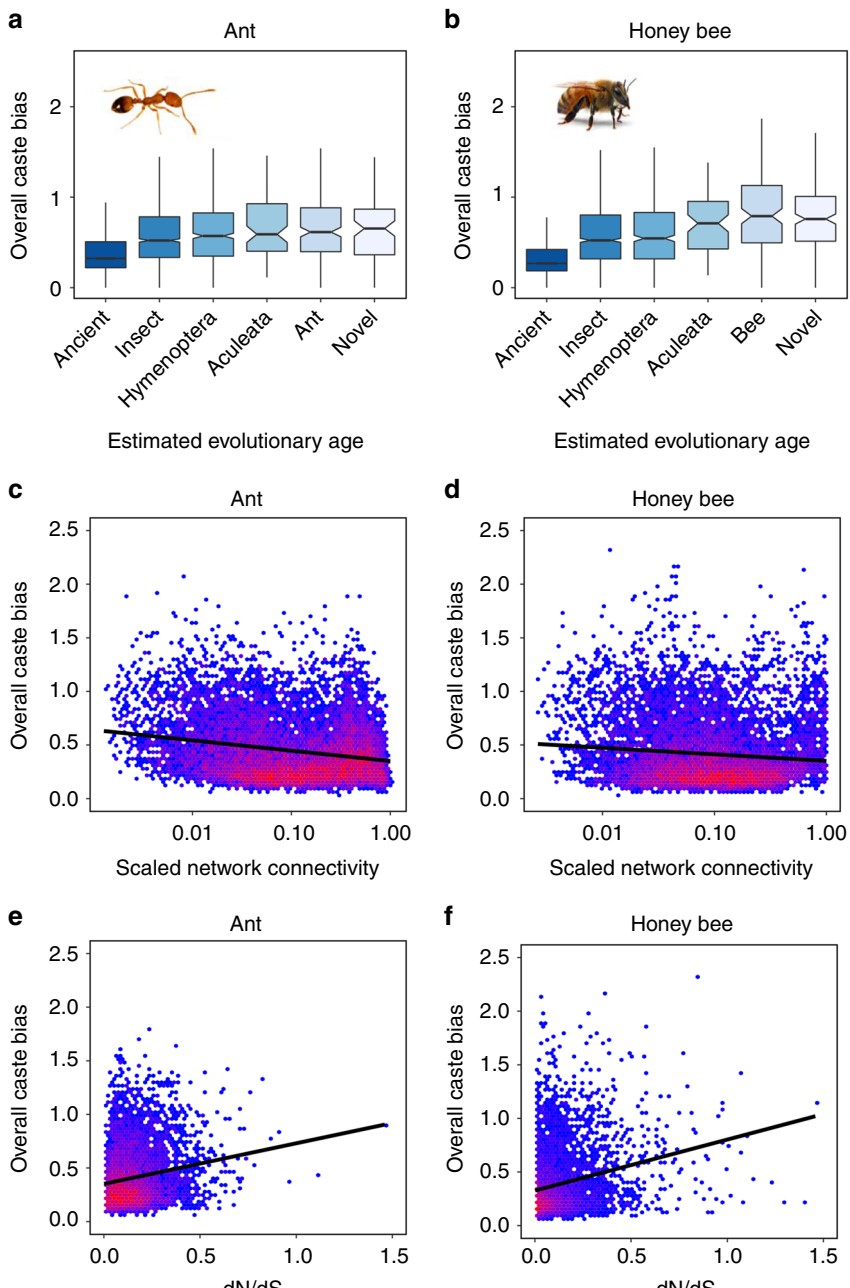

**Fig. 4** Evolutionary and network features of caste-biased genes. Genes that exhibit more caste bias across tissues and developmental stages have younger estimated evolutionary ages (**a**, **b**) and tend to be loosely connected (**c**, **d**; Spearman correlation; ant: rho = −0.159, $P < 0.001$; honey bee: rho = −0.090, $P < 0.001$) and rapidly evolving (**e**, **f**; Spearman correlation; ant: rho = 0.157, $P < 0.001$; honey bee: rho = 0.240, $P < 0.001$). Overall caste bias combines queen/worker $\log_2$ fold-change values across all development stages and adult body segments. Connectivity is calculated using all samples and genes and scaled proportionally to the highest value. In **a** and **b**, middle line represents median values, outer edges of boxplot represent upper and lower quartiles, and whiskers represent a deviation of 1.5*(interquartile range) from the upper and lower quartiles. Source data are provided as a Source Data file. Photos were taken by Luigi Pontieri (pharaoh ant) and Alex Wild (honey bee)

between species (Supplementary Fig. 10a, b), and expression plasticity between queens and workers was correlated to expression plasticity between nurses and foragers (Supplementary Fig. 11). Genes with high levels of caste or behavior bias tended to exhibit a suite of network and evolutionary features including being loosely connected in regulatory networks, evolutionarily young, and rapidly evolving (Fig. 4; Supplementary Fig. 12), as well as displaying tissue-specific expression profiles (Supplementary Fig. 10) in comparison with more ubiquitously expressed genes.

These network and evolutionary characteristics have commonly been implicated for genes underlying eusocial evolution[36,54–56], particularly in association with the worker caste[28–33]. While factors such as evolutionary age and rate to some degree cannot be reliably disentangled[57], these characteristics together reflect relaxed selection on genes' coding sequences and expression profile. This may indicate that caste bias evolves from pre-existing expression plasticity[54]. This could occur when genes that were previously tightly regulated in another context acquire biased expression[58], which is possibly reflected in our

results by the association between tissue specificity and caste/behavior bias in honey bees (Supplementary Fig. 10). Alternatively, caste-biased expression could evolve neutrally, in which genes with loosely regulated expression patterns acquire caste-biased expression randomly, through neutral or slightly deleterious substitutions in regulatory sequences[56,59]. Our results are consistent with both mechanisms for the evolution of caste bias: a large part of abdominal caste bias seems to evolve through the novel regulation of genes with sex-specific expression plasticity, while the bulk of the genes underlying caste-based division of labor exhibit characteristics generally reflective of genes which are weakly constrained.

Our study shows that the recruitment of a large core of conserved reproductive-associated genes, which can be described as a reproductive groundplan, is fundamental to the convergent evolution of caste-based division of labor in ants and honey bees. However, our study also reveals that the bulk of the full genetic architecture underlying the expression of social insect caste-based division of labor varies between lineages. This is reflected by the general biology of social insects, in that independently evolved societies share reproductive division of labor, the main defining feature of eusociality, but also display a wide diversity of lineage-specific adaptations[7]. Future studies including more species will be necessary to determine the generality of the patterns (e.g., the precise numbers of shared and lineage-specific genes) we found. It is likely that a relatively small number of core conserved genes exist as upstream hubs in regulatory networks, and layered downstream of this core is a myriad of taxonomically restricted genes as well as conserved genes with lineage-specific expression patterns[6,32,33,60]. This is consistent with models for the evolution of hierarchical developmental gene-regulatory networks, whereby a relatively small number of highly conserved genes act upstream to initiate gene cascades (e.g., to set up body patterning), while batteries of downstream genes are evolutionarily labile and largely responsible for lineage-specific features[61]. Recent studies have made progress elucidating the function of several core genes and pathways for caste[19,21,22,62]. Large-scale transcriptomic studies such as ours serve a complimentary, indispensable role of identifying the full suite of genes underlying caste-based division of labor in multiple independent lineages.

## Methods

**Study design**. We collected parallel time series RNA-seq data of caste development in the pharaoh ant *Monomorium pharaonis* and the honey bee *Apis mellifera*, including seven developmental stages (egg, five larval stages, one pupal stage) plus each of three adult body segments (head, thorax, abdomen) in both species (Supplementary Table 1). We separated adults into the three main body segments (head, mesosoma, and metosoma) upon sample collection and sequenced pools of each body segment separately. For convenience, we refer to these segments as head, thorax, and abdominal tissues throughout. We sequenced whole embryos and whole bodies of larvae and pupae. We collected three biological replicates of each specific sample type. Each biological replicate contained a pool of individuals ($N =$ 10 for ants, $N = 5$ for honey bees) from the same colony, such that each biological replicate corresponds to a colony. The only exception to this was mature honey bee queens, which were sampled from separate unrelated colonies. In collecting these samples, we complied with all relevant ethical regulations for animal testing and research.

**Ant collection**. To collect samples of *M. pharaonis* across development, we created 27 replicate colonies of ~400 workers and ~400 total larvae from a large mixed genetic source. We removed queens from each colony, which stimulates the production of new queens and males from existing eggs and L1 larvae in *M. pharaonis*[63,64]. We pre-assigned each colony to one of nine sample types, ordered by developmental timing (egg, L1–L5 larvae, pupae, virgin queens/males, nurses/foragers). We allowed the 27 colonies to grow for 4 weeks, and collected samples progressively when the youngest individuals left in the colonies represented the assigned developmental stage (note that *M. pharaonis* workers, lacking ovaries[65], do not begin to lay eggs, so the brood progressively ages as no replacement eggs are laid).

We identified larval stage and caste as previously described, by hair and morphology[66]. To synchronize pupal developmental collection, we exclusively

sampled pupae whose eyes had darkened. We sampled males as soon as they had eclosed as adults from the pupal stage. While *M. pharaonis* does exhibit age polyethism with respect to nursing and foraging[67], the precise dynamics with regard to age are not well studied in comparison with honey bees. Therefore, we distinguished between nurses and foragers based on behavioral observation. Specifically, we observed nurses feeding larvae and we observed foragers collecting food. We sampled egg-laying mature queens (3–4 months old) from the group of queens we initially removed to stimulate reproduction to ensure that queens and workers came from the same genetic background.

**Honey bee collection**. To collect samples of *A. mellifera* across development, we established experimental colonies, in which queens were allowed to lay directly onto empty comb for 24 h to ensure control of larval age and that larvae of a given replicate were from the same queen. We collected eggs after this period directly from the comb. After 3 days, we grafted a subset of hatched larvae into artificial queen cells in queenless portions of the hive. Starting at that day, we sampled the five stages of larvae (L1–L5) on each consecutive day. We sampled pupae once their eyes had darkened to synchronize developmental timing. We sampled males by placing hair-roller cages on top of cells during pupation and waiting for individuals to emerge from pupation. We sampled egg-laying mature queens from separate, unrelated colonies and pooled them into replicate samples. We paint-marked worker individuals upon emergence from the pupal stage and sampled nurses that were less than 7 days old and foragers that were greater than 21 days old.

**RNA extraction, sequencing, aligning to genomes**. We isolated RNA using Trizol reagents. We performed cDNA synthesis and library preparation using a previously described protocol[68], with the only alteration being that the input RNA was 50 ng and the cycle number of cDNA amplification was increased to 16. To compare sample quality across the experiment and test our ability to detect lowly expressed genes, we added ERCC92 (Thermo Fisher Scientific Inc.) spike-in mixes to the total RNA prior to amplification. We pooled libraries with an equal amount of cDNA and sequenced single-end for 50 cycles in Illumina Hiseq 2500. We aligned reads to reference genomes using Bowtie2[69]. All reads were aligned to NCBI gene models (*A. mellifera* genome version 4.5, *M. pharaonis* genome version 2.0, and *D. melanogaster* assembly release "6 plus ISO"). We estimated read count and transcripts per million (TPM) using RSEM[70].

**Differential expression analysis**. To identify caste-associated differentially expressed genes (DEGs), we performed differential expression analysis between queens and workers at each developmental stage and tissue, separately for each species. We removed lowly expressed genes that did not meet one of two criteria: (1) counts per million (CPM) greater than one in at least half the samples, or (2) CPM > 1 in all samples of a given tissue/stage/caste combination (to ensure tissue-specific genes were retained). We removed 2350 lowly expressed genes in ants, leaving 10,804 genes for further analysis, and we removed 2036 genes in bees, leaving 11,775 genes for further analysis. We constructed GLM-like models, including replicate and caste, and identified genes associated with caste at each stage or tissue using EdgeR[71]. Similarly, to identify behavioral DEGs, we performed differential expression analyses between nurses and foragers for each tissue. To identify developmental DEGs in each species, we constructed models with all larval and egg samples and identified genes differentially expressed between any developmental stage, controlling for overall caste differences. To estimate gene-wise sex bias of *D. melanogaster* orthologs, we downloaded available whole-body RNA-seq data[44], consisting of one 5-day-old and one 30-day-old fly of each sex, and performed differential expression analysis as above.

**Identification of orthologs**. To identify orthologs between *A. mellifera* and *M. pharaonis*, we started with a curated orthology map of aculeata species from OrthoDB9[72]. We downloaded amino acid sequences for each species from RefSeq[73]. We associated transcripts with OrthoDB9 protein names using BLASTp (E-value $10^{-10}$) and identified the aculeata ortholog group matched by each gene based on the identified BLASTp hits. In this way, we identified 1:1, one-to-many, and many-to-many orthologous groups between *A. mellifera* and *M. pharaonis*. For direct comparison of the species, we restricted our analysis to 1:1 orthologs (i.e., genes for which only one gene from each species matches the given OrthoDB9 ortholog group). We identified three-way 1:1:1 orthologs between *A. mellifera*, *M. pharaonis*, and *Drosophila melanogaster* using a similar procedure based on endopterygota orthology groups from OrthoDB9.

**Gene co-expression analysis**. In contrast to many network methods which assess gene–gene relationships across all samples, biclustering seeks to identify a group of genes which are coexpressed (i.e., exhibit concerted expression changes) across a subset of sample types[74]. Given that our data contained a large number of sample types, we reasoned that we could employ biclustering to identify groups of genes particularly associated with a given sample type. While our level of biological replication ($N = 3$ for each tissue/caste/stage combination) is low, including all samples in our biclustering analysis allows high resolution of gene–gene co-expression relationships, and biclustering allows for specificity of gene–sample relationships. We performed plaid clustering, one of the top performing

biclustering algorithms in a recent survey[75], using the R package biclust[76]. Plaid clustering models expression level for each gene as a function of bicluster weights, where only biclusters containing the gene contribute to predicted expression level[77]. The algorithm iteratively constructs layers containing samples and genes and retains layers that improve the model fit, where layers represent biclusters.

Plaid clustering is non-deterministic, and individual biclusters are not found in every iteration of clustering. To define a reasonable ensemble of biclusters, we performed clustering 1000 times separately for each species, using inverse hyperbolic sine transformed TPM (transcripts per million)[78]. While a large number of interesting bicluster definitions are possible, we decided to identify biclusters that consistently contained all queen-abdomen samples to focus our investigation on the tissue that exhibited the strongest signature of caste bias. Specifically, we extracted biclusters containing all three mature queen-abdomen samples and no more than three other samples total. Honey bee queen-abdomen samples clustered with egg samples, while pharaoh ant queen samples did not cluster with egg samples. It is possible that this difference is a result of a difference of age of the eggs at time of collection: honey bee eggs were 24 h old and likely still contained maternal RNA, while pharaoh ant eggs were 7 days old.

Because the same genes were not always present in such a bicluster, we tabulated the number of queen-abdomen biclusters each gene was found in and retained genes present in a higher proportion of biclusters than a given cutoff, determined by inspection of frequency distributions of bicluster presence. In pharaoh ants, we found a large set of genes present in greater than 90% of queen-abdomen biclusters, and we retained these genes for further analysis ($N = 1006$ genes; Supplementary Fig. 14a, i.e., the same set of genes was repeatedly found). In contrast, honey bee queen-abdomen biclusters tended to contain one of two groups of genes, as the frequency of presence in the bicluster peaks at 60 and 30% (Supplementary Fig. 14b). Out of 1174 genes present in greater than 60% of the identified biclusters, 877 were differentially expressed and upregulated in queen abdomens relative to worker abdomens (also note that this set of genes exhibited much higher expression in eggs than the latter set). In contrast, out of 1057 genes present in 25–35% of biclusters, 611 out of were differentially expressed and upregulated in worker abdomens, compared with 47 upregulated in queen abdomens. Therefore, it is clear that the more common bicluster represents genes associated with queen abdomens, so we retained this set of genes for further analysis ($N = 1174$ genes).

We proceeded with our analysis using these identified sets of genes, which we term modules associated with queen abdominal expression. We calculated connectivity in the module (i.e., intramodule connectivity) as the sum of pairwise Pearson correlations, where correlation values are raised to the sixth power, the standard value for unsigned weighted gene co-expression networks[79] (note that we calculated total connectivity, used in Fig. 4, across the entire network using all samples and all genes). A major goal of gene co-expression analysis is the identification of hub genes, genes central to networks that are strongly associated to relevant traits[45]. To this end, we conservatively identified hub genes associated with queen abdominal expression as genes with intramodule connectivity in at least the 90th percentile and abdominal $\log_2$ fold-change values greater than two (representing a fourfold increase in expression in queen relative to worker abdomens).

**Phylostratigraphy**. We estimated the evolutionary age of each gene using phylostratigraphy. Phylostratigraphy groups genes into hierarchical age categories based on identifiable orthology (using BLASTp)[80,81]. For example, genes found in ants and honey bees but not in non-aculeate hymenopterans would be labeled "aculeata" genes, while genes shared between vertebrates and insects would be labeled "bilateria". For our purposes, we decided to focus on the difference between "ancient" genes, which we defined as displaying orthology with non-insect animals, and a number of hierarchical younger categories: insect, hymenopteran, aculeate, ant, bee, and novel (where "ant" refers to genes found in *M. pharaonis* and other ants but not in any other species, "bee" refers to genes found in *A. mellifera* and other bees but not in other species, and "novel" refers to a gene found only in *A. mellifera* or *M. pharaonis*).

A key component of phylostratigraphy is the creation of a BLAST database in which to identify orthologs[80,81]. Because we largely planned to focus on younger age categories, we constructed a protein database containing all annotated hymenopteran genomes (48 total). We added to this group ten non-hymenopteran insect genomes and ten non-arthropod genomes (see Supplementary Table 12 for a full list of included genomes). Therefore, a gene labeled as "ancient" displayed a significant BLASTp hit to one of the ten non-arthropod genomes. While phylostratigraphy typically employs an extremely large database containing all available representative taxa, we reasoned that for our study resolution between categories, such as bilateria and eukaryota was unnecessary. Furthermore, adding extraneous genomes effectively dilutes the database, such that more similarity is needed to pass an E-value threshold. Because we included only a sample of non-hymenopteran genomes, we were therefore able to stringently identify orthologs (E-value $10^{-10}$ in comparison with a typical value of $10^{-5}$)[81] and accurately place them along the hymenopteran phylogeny.

**Estimation of tissue specificity**. We downloaded available RNA-sequencing data on 12 tissues in *A. mellifera* worker nurses and foragers[32]. To classify genes by their tissue specificity, we calculated $\tau$, a commonly used metric of expression specificity[82]. $\tau$ ranges from 0 to 1, where 0 indicates that genes are ubiquitously expressed and 1 indicates that genes are exclusively expressed in one tissue.

**Estimation of evolutionary rate**. We estimated evolutionary rate using dN/dS, the ratio of non-synonymous to synonymous nucleotide changes. We estimated pairwise dN/dS between each focal species and a second closely related species with an available genome (*A. mellifera*:*A. cerana*; *M. pharaonis*:*S. invicta*). For each 1:1 ortholog pair, we selected the longest transcript associated with the gene for each pair of species. We aligned orthologous protein sequences using ClustalW[83], derived nucleotide alignments from protein alignments using pal2nal[84], and estimated pairwise dN/dS of nucleotide alignments using PAML, package codeml[85].

**Partial correlation analysis**. We performed partial Spearman correlations between overall bias and evolutionary/network characteristics, controlling for the effect of expression.

**Gene ontology analysis**. We performed Gene Set Enrichment Analysis (GSEA) using the R package topGO[86]. We utilized the well-curated *D. melanogaster* gene ontology database, downloaded from FlyBase[49]. We performed GSEA analysis on genes with 1:1:1 orthologs, associating the *D. melanogaster* Gene Ontology (GO) terms to *A. mellifera* and *M. pharaonis* orthologs. We identified GO terms associated with caste- or behavior biased differentially expressed genes using the P-value of differential expression between queens and workers or nurses and foragers. We identified GO terms associated with overall caste or behavior bias using the Euclidean distance of $\log_2$ fold change between queens and workers or nurses and foragers at each stage. We identified enriched terms with P-value < 0.05.

**Reporting summary**. Further information on research design is available in the Nature Research Reporting Summary linked to this article.

## Data availability
All data and scripts required to generate figures, tables, and perform statistical analyses are available on Github: https://github.com/warnerm/devnetwork. Raw reads are deposited at NCBI SRA, Bioproject PRJNA533756. The source data underlying all figures are provided as a Source Data file.

## Code availability
All scripts required to perform all analyses and generate figures and tables are available on Github: https://github.com/warnerm/devnetwork.

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

## Acknowledgements

We would like to thank the following: Luigi Pontieri for photographs of pharaoh ants, Alex Wild for photographs of honey bees, Chao Tong for compiling hymenopteran genomes for use in phylostratigraphy and for comments on the manuscript, Rohini Singh for comments on the manuscript, and Junhyong Kim, Mia Levine, and Justin Walsh for helpful comments and discussion. This work was funded by the National Science Foundation (grant number IOS-1452520 to TAL) and United States Department of Agriculture National Institute of Food and Agriculture (grant number 2014-67013-21725 to TAL), and subsidy funding from Okinawa Institute of Technology to ASM. The National Science Foundation also funded MRW (DGE-1321851).

## Author contributions

M.R.W., T.A.L. and A.S.M. designed the study. M.R.W. and M.J.H. collected samples. L.Q. performed all RNA extractions, library preparation, and sequencing. A.S.M. performed all bioinformatic analyses to process, filter, and align *A. mellifera* and *M. pharaonis* RNA-sequencing reads to genomes. M.R.W. performed all other transcriptomic, comparative genomic and subsequent analyses, with input from T.A.L. and A.S.M. M.R.W. and T.A.L. wrote the paper, with input from A.S.M.

## Additional information

**Competing interests:** The authors declare no competing interests.

