## [Peer Review File · Nature Communications]

Reviewers' Comments:

Reviewer #1:

Remarks to the Author:

Warner et al. use comparative transcriptomics to explore the genomic basis of reproductive and behavioral castes in eusocial insects, an example of convergent evolution. The authors describe caste- and tissue-specific gene expression profiles for two species representing two independent origins of eusociality. They find evidence both for a shared core of genes regulating caste in these two species, as well as many lineage-specific genes. This study is of interest not only to those who study social behavior, but also evolutionary biologists interested in the molecular basis of convergent complex phenotypes.

A nice feature of this study is carefully coordinated studies across different species

While I think this is an important study and don't disagree with the main conclusions of the paper, I have one main criticism that affects the interpretation and presentation of some of the results. The authors suggest that caste-biased genes are derived from sex-biased genes based on a correlation between log₂ fold changes between queen and worker abdomens and between female and male *Drosophila melanogaster* samples. I can see some reasons why they would see this correlation that does not result from a derivation of caste-related genes from sex-related genes. Of primary importance is that the signal coming from queen and female samples in the three species reflects a signal of ovary activation and active reproduction, which is not sexual dimorphism. Unmated, non-reproductive *Drosophila* females are still female, yet would likely show a very different expression pattern relative to males than mated females (as I assume were used in the dataset cited, although it's not clear from the text which ENCODE dataset was used). This is especially important because the authors only included adult abdomens in their comparison with sex bias, and in honey bees they found that queen abdomen gene expression clustered with egg gene expression in some modules of coexpression. This means that much of the signal is coming from oocytes, and as such correlations between queen-worker and female-male differences are really more of an egg-no egg comparison than one of sexual dimorphism. The authors could potentially strengthen their argument that caste-related genes are derived from sex-biased genes by incorporating more tissues (not limited to abdomen), and by using the phylostatigraphy analyses to address whether caste-biased genes are older than non-caste-biased genes in general (as you'd expect if they were derived from sex biases). In addition, I have a number of minor comments and editing suggestions as follows:

Line 16: Eusociality is traditionally characterized by more than just caste-based division of labor

Line 20: In abstract but also throughout, authors give no justification for their choice of species

Line 22: What is relevance of upregulation in female flies? Are these mated, reproductively active flies? Is the point that queens show "typical" reproductive-related gene expression relative to non-social insects?

Line 23: The authors refer to the shared core of genes as "large", but then say the majority of genes are not commonly differentially expressed across the two species. This language seems at odds

Line 25: evolutionary should be evolutionarily

Line 46: Age polyethism is not present in workers of all eusocial insects. A description of the biology of the two species used would be helpful to clarify that in these two species, age polyethism occurs, but perhaps also a note recognizing that this is not the case in all species would be helpful to non-specialists.

Line 96: Just to clarify, RNA-sequencing libraries were enriched for mRNA and did not include miRNAs and other small RNAs?

Line 104-105: Text seems inconsistent with Figure- in ant abdomen, there appear to be more shared than unshared caste-associated DEGs

Line 105: I don't see how this sentence beginning with "Similarly" is a similar statement to the one before

Line 109: typo: "N=7640 1:1 orthologs" is repeated

Line 111: Not clear which "caste-associated DEGs" were used in the GO enrichment- shared across species, not shared, DEGs in either species? (From supplemental tables it's clear that GO was done on each species separately, but needs to be clear in main text.)

Line 121: Same question as on line 111

Lines 136-137: "genes varied in expression bias across evolutionary age categories" – meaning unclear

Line 143: clarification- egg and larval development only?

Line 147 (and surround section): Why the focus on queen abdominal expression? Why not also look at modules associated with worker-upregulated genes?

Line 151: missing word "with" after "associated"

Line 152: "clearly associated with reproduction and maternal effects"- how is it clear? By eye? I'd like more evidence, such as GO enrichment of hub genes. If the list of hub genes is too small to perform GO analyses, the authors might consider using a rank-based approach with degree of connectivity for GO enrichment

Line 161: a stretch to suggest hub genes are the "most important"?

Line 162-163: I don't follow the logic- why would caste-biased expression be derived from sex-biased expression because queen-biased genes are related to reproduction? Queen-biased genes are likely related to reproduction because queens are reproductively active... not sure why that would suggest anything about a relationship between caste and sex.

Line 163: Not clear where the sex-biased expression data come from- no mention of males up to this point

Line 163-165: Again, I don't agree that a correlation between queen-worker logFC and queen-male logFC indicates a derivation of caste-biased expression from sex-biased expression. Queen vs. not-queen will give DEGs related to differences in reproduction, whether the comparison is against workers, males, virgin queens, etc. The logFC in both cases could be driven by queen-related expression alone. Even without that, this correlation doesn't indicate direction- couldn't strong selection on caste differences lead to constraints on gene expression breadth leading to the observed correlation?

Line 168: typo: "effect" should be "effects"

Line 175: Similar to comment above, worker-biased abdominal genes showing upregulation in males could simply be lack of downregulation related to ovary development in both groups; not necessarily related to sex differences per se

Line 193: the fact that plasticity in gene expression is correlated across contexts, as noted by the authors, is the very reason I'm skeptical of the interpretation of the correlation between caste-biased expression and sex-biased expression

Line 209: missing word "of" between "values" and "caste"

Line 211: to my knowledge ref 31 has honey bee data only; does this mean the expression tissue specificity was calculated based on honey bee expression only, not also for ants?

Line 222: "many genes with known roles in reproduction"- again I'd like to see something more formal to assess whether reproductive-related genes are enriched in the shared set- GO or something similar

Line 223-224: "seems to be derived from ancient plastically-expressed genes underlying sexual dimorphism"- I don't buy this. Just because there's overlap in female *D. melanogaster* relative to male DEGs with caste-biased genes doesn't mean caste genes are derived from sex genes. Are the *D. melanogaster* DEGs really representative of sexual dimorphism? How can correlation between logFCs in two studies imply direction of evolution?

Line 227: the authors indicate that no previous comparative study has investigated caste-biased expression in the abdomen. While it's true that no studies have sequenced multiple species in the same study, there are multiple studies of abdominal queen-worker gene expression differences, many of which have made comparisons across studies post-hoc. While these comparisons are less ideal than the current study due to technical differences, as noted, the authors could make an attempt to include at least references to this work in the discussion, and potentially look for hub genes and other

overlapping "important" DEGs from this study in those, particularly for additional independent lineages of eusocial insects.

Line 242-244: Perhaps it's not surprising that honey bees and pharaoh ants show little overlap in nurse-forager DEGs- it is known that honey bees show different relationships of some of the canonical signaling pathways involved in division of labor (such as the relationship between JH and Vg). The authors could comment on this

Line 255: missing word after "In contrast"

Line 266: typo, word "of"

Line 268: "many traits"- do these traits share any particular features in common that make them different from morphological traits? Specific to behavior? Complex traits? Polygenic traits?

Line 281: "on top" is semantically confusing, as I believe the authors are referring to TRGs that are downstream in the network

Fig. 2A and B: blue worker-upregulated points difficult to see

Fig 3: "Caste bias is derived from sex bias" – the correlation shown does not convince me of this

Supplement line 44: extra words- "were performed"

Supplement line 51: missing end parenthesis

Supplement line 56: Aculeata should not be italicized

Supplement line 84-85: can the authors comment on why honey bee egg samples would cluster with queen abdomen samples? This suggests to me that the abdominal gene expression signal is coming primarily from oocytes in the ovaries.

Supplement lines 93 and 96: gene numbers (1039, 1245) are not consistent with those in the main text (lines 149-150, 1006, 1174)

Supplement line 134-135: what is "value" of overall caste bias or overall behavior bias? logFC? Is p-value uncorrected or corrected for multiple testing?

Fig. S3 legend- typo, "indicating" should be "indicate"

Fig. S3- please indicate which stage/tissues have significant differences in logFC across phylostrate, because it is not given in main text or figure which (if any) are significant

Fig. S4- clarification needed about "between developmental stages"- this is referring only to larval developmental stages, correct?

Reviewer #2:

Remarks to the Author:

The authors set out to understand the genetic underpinnings of caste determination in social insects and to determine whether or not species that represent independent origins of eusociality share genetic mechanisms underpinning division of labor. They did this by examining tissue-specific patterns of gene expression in pharaoh ants and honey bees, which represent two different origins of eusociality. They compared the expression profiles of several different developmental timepoints and focal tissues in queens vs workers, nurses vs foragers and males.

I am very excited to finally see a study that captures a much more complete range of tissues and timepoints in two different species. It is about time someone did this! However, I was surprised that several confounding factors may not have been accounted for in the study design and were not discussed in the manuscript.

The main findings the authors emphasize in the study are that: (1) a large proportion (~30%) of the genes differentially expressed between queens and workers in ants and bees are shared, (2) these genes are hub-like with higher levels of connectivity in co-expression networks, (3) these differences

are correlated with differences in sex-biased expression, and (4) that these genes are also associated with reproduction in *Drosophila*.

Major comments:

To what extent are these differences just driven by the presence of developing eggs inside reproductively-active queens vs non-reproductive workers and males? In other words, how much of this signal is driven by egg expression alone? Developmental genes presumably being expressed in the embryos present inside the queens' ovaries are already known to exhibit all of the above properties and have older evolutionary origins. In fact, the authors find that (at least in the honey bees), many of the queen abdominal samples cluster with egg samples in their coexpression analyses. Because of this, it is unclear to me how much of the findings and results presented in this study are actually driven by caste-based differences between queens and workers and relevant to the evolution of eusociality. This would be more compelling if this authors could also demonstrate this was true in other tissue types (such as the brain). However, in all other tissues, this pattern does not appear to hold.

Instead it seems to be younger, taxon-restricted genes that are differentially expressed, and these differences appear to be unique to each species examined. This, in my opinion, is the most interesting finding in this study. I was bit surprised that this finding was less emphasized than the shared abdominal expression patterns. However, I appreciate the difficulties associated with proving a negative result and the difficulties of making functional comparisons across different species.

It was unclear from the methods whether the authors took any steps to minimize differences associated with the specific life histories of each species. For example, how comparable are behavioral classes across each species? I believe both have an age-based division of labor, but are these dynamics similar in both? For example, are only foragers exposed to light in both species?

Did the authors make any attempt to control for age-associated differences between nurses and foragers? Many of the behavioral DEGs were associated with development and metabolism, and age can have large effects on these functional groups as well. Similarly, it seems males were only 1 day old; how might this affect the results?

Minor comments:

More details on how nurse/forager classifications were determined would be helpful. Was it just an age-based categorization, or were more detailed behavioral observations made to confirm these behaviors?

More details on the specifics of this analysis of the DEG analysis would be very helpful

- It seems the primary data filtration was a minimum CPM < 1 in all samples. Was there a minimum # of samples that had to have CPM>1?
- How many of the DEGs had low expression levels?
- Was there a minimum fold change required to consider something differentially expressed?

Similarly for the co-expression and clustering analyses, more methodological details would have been useful.

For the co-expression analyses, what happens when you use the same cutoff for honey bees and ants? Please clarify why two different cutoffs were selected for each species.

Please specify the numbers of colonies and individuals for honey bees /ants for each behavioral class; were individuals related or not, especially with respect to queens?

Reviewer #3:

Remarks to the Author:

The authors generated queen and worker RNA-seq data for a pharaoh ant and a honey bee species, generated from multiple larval instars, pupae, and adult body segments. For adult workers, they also generated separate transcriptomes from adult nursing workers and foraging workers. The parallel use of common methods and a common sampling scheme represents an important advance over prior meta analyses of castes in ants and bees. Biological replication is somewhat modest, with 3 replicates for each sample type, but the sheer number of sample types considered in the study is an asset, and the statistical analyses and inferences appear sound. Remarkably little RNA-seq data exists that compares queen and worker gene expression in the honey bee *A. mellifera*, and I anticipate this developmental time course data will be of value to future studies, particularly given the importance of *A. mellifera* as a model eusocial insect. The manuscript is framed in terms of investigating the extent of molecular convergence exhibited by independently-evolved caste systems in a bee and an ant. As emphasized in the title, the authors observe that many genes fundamental to female reproduction are associated with reproductive division of labor, but they also reconcile this with the existence of much higher numbers of lineage-specific, plastic genes. I find that this treatment of the findings provides useful insight and perspective on hypotheses that have served as a hotbed of recent research by the social insect community. Several suggestions for revisions follow.

L96: Specify level of biological replication ($n = 3$) in main text – here or in methods.

The authors used an FDR corrected P-value threshold of < 0.1 to generate very large lists of differentially expressed genes. I'm curious if the trends related to DEG categorization in Figure 1 are robust to more stringent cutoffs. Does the use of FDR < 0.05 or a combined FDR + fold-change cutoff alter any of the findings related to detectable orthology or phylostrata in Figure 1?

Although there are many sample types, the use of 3 biological replicates seems low for a gene coexpression analysis, particularly given that the network module for queen abdominal expression is largely informed by a comparison of 1 sample type (adult queen abdomen) to the others. Can the authors comment on this issue? Similarly, can the same insights from Figure 2 be provided by analyses of gene expression levels alone (e.g., are DEGs present in both species more likely to have larger $\log_2(Q/W \text{ abdominal})$ values than DEGs in only one species)?

L190: please provide correlation coefficients in text.

Results from Figure 4 showing that degree of gene expression bias is positively correlated with evolutionary rate may warrant citing Hunt et al. 2011 (<https://doi.org/10.1073/pnas.1104825108>) in the discussion given the similarities with these prior findings.

L265: I don't think theory from either ref 49 or 51 supports an expectation that most gene expression differences among morphs would be found in highly conserved hub genes rather than downstream peripheral genes. Can this part of the discussion be toned down or made more clear?

Reviewer 1

1. Warner et al. use comparative transcriptomics to explore the genomic basis of reproductive and behavioral castes in eusocial insects, an example of convergent evolution. The authors describe caste and tissue-specific gene expression profiles for two species representing two independent origins of eusociality. They find evidence both for a shared core of genes regulating caste in these two species, as well as many lineage-specific genes. This study is of interest not only to those who study social behavior, but also evolutionary biologists interested in the molecular basis of convergent complex phenotypes.

A nice feature of this study is carefully coordinated studies across different species

Thank you very much for these positive comments, we agree that our study is of broad interest.

2. While I think this is an important study and don't disagree with the main conclusions of the paper, I have one main criticism that affects the interpretation and presentation of some of the results. The authors suggest that caste-biased genes are derived from sex-biased genes based on a correlation between log₂ fold changes between queen and worker abdomens and between female and male *Drosophila melanogaster* samples. I can see some reasons why they would see this correlation that does not result from a derivation of caste-related genes from sex-related genes.

Of primary importance is that the signal coming from queen and female samples in the three species reflects a signal of ovary activation and active reproduction, which is not sexual dimorphism. Unmated, nonreproductive *Drosophila* females are still female, yet would likely show a very different expression pattern relative to males than mated females (as I assume were used in the dataset cited, although it's not clear from the text which ENCODE dataset was used).

Thank you for these important and careful comments. We agree that the main signal we are detecting is likely associated with *active* female reproductive physiology (i.e. transcriptomic patterns associated with egg production). This is not sexual dimorphism in the sense of external morphological differences between the sexes, however, it is still sexual dimorphism for reproductive function. Such a signature of reproductive function is only detectable in reproductively active individuals, so we agree that non-reproductive social insects and flies would not show this signature. In the revised text, we clarified that the *D. melanogaster* samples came from mated females and males (L: 22) and we also further clarified that our results are most consistent with the genes underlying the queen reproductive caste being recruited from highly conserved genes underlying female reproduction (L: 244-245)

3. This is especially important because the authors only included adult abdomens in their comparison with sex bias, and in honey bees they found that queen abdomen gene expression clustered with egg gene expression in some modules of coexpression. This means that much of the signal is coming from oocytes, and as such correlations between queenworker and female-male differences are really more of an egg-no egg comparison than one of sexual dimorphism. The authors could potentially strengthen their argument that caste-related genes are derived from sex-biased genes by incorporating more tissues (not limited to abdomen), and by using the phylostratigraphy analyses to address whether caste-biased genes are older than non-caste-biased genes in general (as you'd expect if they were derived from sex biases).

Thank you for these suggested additional analyses. As suggested, we repeated our comparison of *D. melanogaster* sex-bias to shared caste-bias, with caste-bias defined by comparing queen and worker head or queen and worker thorax. While we identified few shared caste-associated DEGs in each tissue (N = 38 [head] and N = 64 [thorax]), these DEGs showed the same pattern as we observed in abdominal tissue -- genes with conserved queen-bias were more female-biased in *D. melanogaster* than genes with shared worker-bias. We added a supplemental figure to show this result (Fig. S8) Additionally, we detected a correlation between caste- and sex-bias within each social insect species (Fig. S7). We added (L: 257-263) to the Discussion to explain these conclusions, and our interpretation that these findings mean that the association of caste- and sex-bias is not simply due to the presence of oocytes but rather is more general and associated with female reproductive physiology. That said, as we explain above, and as you point out, this is

distinct from basic sexual dimorphism (e.g., for morphology) that would be found between all males and females, regardless of reproductive status.

In addition, I have a number of minor comments and editing suggestions as follows:

4. Line 16: Eusociality is traditionally characterized by more than just caste-based division of labor

We agree that eusociality has long been defined by three features: cooperative brood care, overlapping generations, and reproductive division of labor. We believe that all three of these features fit into the phrase we used, “caste-based division of labor”: the worker caste participates in cooperative brood care, the presence of worker offspring at the same time as colony queen(s) means that there are overlapping generations in the colony, and the caste system defines the reproductive division of labor. However, to be clear we changed the sentence to: “Eusociality has convergently evolved multiple times, but the genomic basis of caste-based division of labor and degree to which independent origins of eusociality have utilized common genes remains largely unknown.” (L: 16-18)

5. Line 20: In abstract but also throughout, authors give no justification for their choice of species

Line 20-21 we explain that our study species represent “two independent origins of eusociality”. Similarly, lines 85-87 we explain that our two study species “represent two independent origins and elaborations of eusociality”. To further elaborate, we have revised the main text by adding a new sentence stating “We chose these two study species because they represent two independent origins of eusociality in the ant and corbiculate bee lineages⁴¹ as well as two independent elaborations of eusociality, each characterized by strong queen-worker dimorphism and age-based worker division of labor^{32,42}” (L: 83-87)

6. Line 22: What is relevance of upregulation in female flies? Are these mated, reproductively active flies? Is the point that queens show “typical” reproductive-related gene expression relative to nonsocial insects?

Yes, the point is that queens show a general reproductive-related physiology, based on gene expression profile. Thank you for suggesting this clarification, we have changed the text to “...upregulated in mated female flies, indicating that these genes are part of a conserved insect reproductive groundplan.” (L: 22-23)

7. Line 23: The authors refer to the shared core of genes as “large”, but then say the majority of genes are not commonly differentially expressed across the two species. This language seems at odds

We agree it does seem contradictory on face value, but we would argue that both are true- 1500 genes are shared, which is a large number, but the majority (65% in pharaoh ants, 71% in honey bees) aren't shared. To further clarify this issue, we have changed “large” to “substantial” (L: 23)

8. Line 25: evolutionary should be evolutionarily

Thank you, we have corrected this mistake.

9. Line 46: Age polyethism is not present in workers of all eusocial insects. A description of the biology of the two species used would be helpful to clarify that in these two species, age polyethism occurs, but perhaps also a note recognizing that this is not the case in all species would be helpful to non-specialists.

Thank you, this is a good point. We have added the word “often” (L: 46) to indicate age polyethism is not ubiquitous, and added a sentence in results (L: 116-118) stating that age polyethism exists in both pharaoh ants and honey bees: “Both honey bees⁴² and pharaoh ants³² exhibit age-based worker division of labor, in which younger individuals tend to specialize on nursing and other within-nest activities and older individuals specialize on foraging.”

10. Line 96: Just to clarify, RNA-sequencing libraries were enriched for mRNA and did not include miRNAs and other small RNAs?

Yes, libraries were enriched for mRNA and did not include miRNA or other small RNAs. Thank you for the suggestion, we have changed text to “mRNA” (L: 96)

11. Line 104-105: Text seems inconsistent with Figure in ant abdomen, there appear to be more shared than unshared caste associated DEGs

We agree this is somewhat unclear. In Figure 1a, the dark blue represents DEGs that are shared between species. The middle blue color represents genes for which orthologs exist but aren't differentially expressed in the other species, while the lightest color represents genes with no ortholog. When all caste-associated DEGs are considered, 35% are shared in ant abdomens. We believe the confusion stems from the difference between the “no ortholog” and “not shared caste-bias” categories, both of which are not caste-biased in the other species. To make this more clear, we have changed the text to “In all tissues and stages, the majority of caste-associated

DEGs in one species were either not differentially expressed or did not have an ortholog in the other species” (L: 104-105)

12. Line 105: I don't see how this sentence beginning with “Similarly” is a similar statement to the one before

We agree and have corrected this issue by removing “similarly”

13. Line 109: typo: “N=7640 1:1 orthologs” is repeated

Fixed, thank you

14. Line 111: Not clear which “caste-associated DEGs” were used in the GO enrichment shared across species, not shared, DEGs in either species? (From supplemental tables it's clear that GO was done on each species separately, but needs to be clear in main text.)

Thank you for pointing this out. We have changed the text to “...for caste-associated DEGs in each species were dominated...” (L: 112) to emphasize that GO analysis was performed in each species separately.

15. Line 121: Same question as on line 111

Changed, same as line 111 (L: 125)

16. Lines 136-137: “genes varied in expression bias across evolutionary age categories” – meaning unclear

Changed to “In general, the evolutionary age of genes was associated with expression bias between castes” (L: 141-142)

17. Line 143: clarification egg and larval development only?

Changed prior description to “embryonic and larval development” (L: 146)

18. Line 147 (and surround section): Why the focus on queen abdominal expression? Why not also look at modules associated with worker-upregulated genes?

While in principle both analyses could be done, we opted to specifically target queen-associated modules to contextualize the large signal of shared queen-upregulation (given that most shared

DEGs were upregulated in queens - 56% compared to 22% that were worker-upregulated in both species). To clarify, we added “We focused on modules specifically associated with queens because the majority of shared DEGs were queen-upregulated.” (L: 153-154)

19. Line 151: missing word “with” after “associated”

Fixed, thank you

20. Line 152: “clearly associated with reproduction and maternal effects” how is it clear? By eye? I’d like more evidence, such as GO enrichment of hub genes. If the list of hub genes is too small to perform GO analyses, the authors might consider using a rank-based approach with degree of connectivity for GO enrichment

We agree that more concrete evidence would be helpful here. Unfortunately, the list of hub genes is indeed too small for GO enrichment analysis, though several of the annotated (and mentioned) genes have known reproductive function. However, we performed a similar supplementary analysis. We identified genes which are known to be associated with oogenesis in *Drosophila melanogaster* based on FlyBase annotations. We tested the relationship between oogenesis association and connectivity within the abdominal module and found that genes associated with oogenesis had higher connectivity than genes not associated with oogenesis in the case of honey bees (Supplementary Fig. 6). We added the text: “Furthermore, genes for which *Drosophila melanogaster* orthologs are known to function in oogenesis (based on FlyBase Gene Ontology⁴⁸) were more highly connected within the queen abdominal modules than genes not associated with oogenesis (Supplementary Fig. 6) for honey bees (Wilcoxon test; $N = 649$; $P < 0.001$), though not for ants ($N = 542$; $P = 0.114$)” (L: 161-166). We also changed the wording “clearly associate..” to “inferred to have functions associated with reproduction and maternal effects” (L: 159) to be a bit more conservative.

21. Line 161: a stretch to suggest hub genes are the “most important”?

We agree, this is a bit speculative. We removed the word “most” (L: 172)

22. Line 162-163: I don’t follow the logic why would caste-biased expression be derived from sex-biased expression because queen-biased genes are related to reproduction? Queen-biased genes are likely related to reproduction because queens are reproductively active... not sure why that would suggest anything about a relationship between caste and sex.

Yes, caste-biased expression should be linked to sex-biased expression because the signal we pick up is a result of female reproductive activity. To be a bit more conservative, we have

changed wording in results to “we reasoned that caste-biased expression would be linked to sex-biased expression” (L: 174)

23. Line 163: Not clear where the sex-biased expression data come from no mention of males up to this point

Added clarification to beginning of results: “adult tissues separated by reproductive caste (queens versus workers), behavior (nurse workers versus forager workers), and sex (queens and workers versus males)” (L: 94-96)

24. Line 163-165: Again, I don’t agree that a correlation between queen-worker logFC and queen-male logFC indicates a derivation of caste-biased expression from sex-biased expression. Queen vs. not-queen will give DEGs related to differences in reproduction, whether the comparison is against workers, males, virgin queens, etc. The logFC in both cases could be driven by queen-related expression alone. Even without that, this correlation doesn’t indicate direction couldn’t strong selection on caste differences lead to constraints on gene expression breadth leading to the observed correlation?

We believe this correlation is the result of a shared signal of female reproductive activity, which predates the evolution of caste. However, we have changed wording in results to “we reasoned that caste-biased expression would be linked to sex-biased expression” (L: 174) to be a bit more conservative

25. Line 168: typo: “effect” should be “effects”

Fixed, thank you

26. Line 175: Similar to comment above, worker-biased abdominal genes showing upregulation in males could simply be lack of down-regulation related to ovary development in both groups; not necessarily related to sex differences per se

Added “, indicative of shared queen (social insects) and female (fly) down-regulation” (L: 190-191). See also our answer to comment #2, where we explain that differences in active reproductive physiology are a type of sex differences.

27. Line 193: the fact that plasticity in gene expression is correlated across contexts, as noted by the authors, is the very reason I’m skeptical of the interpretation of the correlation between caste-biased expression and sex-biased expression

We agree that within species, this correlation could be a result of general expression plasticity. However, our results in comparison to *D. melanogaster* suggest that caste-biased expression is indeed meaningfully associated with (and we would argue derived from) conserved sex-biased expression.

28. Line 209: missing word “of” between “values” and “caste”

Fixed, thank you

29. Line 211: to my knowledge ref 31 has honey bee data only; does this mean the expression tissue specificity was calculated based on honey bee expression only, not also for ants?

Yes, this is correct that ref 31 contains only honey bee data. No comparable RNA-sequencing dataset for several tissues exists in ants. We added clarification that the correlation was tested and present in honey bees “..caste and behavior bias in honey bees compared to more pleiotropic...” (L: 231)

30. Line 222: “many genes with known roles in reproduction” again I’d like to see something more formal to assess whether reproductive-related genes are enriched in the shared set GO or something similar

Yes we agree something more formal is helpful. We added Supplementary Fig. 6 (see answer to comment #20).

31. Line 223-224: “seems to be derived from ancient plastically-expressed genes underlying sexual dimorphism” I don’t buy this. Just because there’s overlap in female *D. melanogaster* relative to male DEGs with caste-biased genes doesn’t mean caste genes are derived from sex genes. Are the *D. melanogaster* DEGs really representative of sexual dimorphism? How can correlation between logFCs in two studies imply direction of evolution?

We agree that interpretation of patterns of evolution based on comparison of transcriptomic studies of three extant species (*M. pharaonis*, *A. mellifera*, and *D. melanogaster*) is difficult. Ideally, transcriptomic ancestral states (i.e. logFCs for males and females) could be inferred based on transcriptomic data spanning the full diversity of extant insects, along with other arthropods. Then, we could quantify the overlap of the inferred ancestral logFC between males and females and our observed logFC between queens and workers in pharaoh ants and honey bees. As this is beyond the scope of our study, we are making the assumption that shared ancestry is the main cause of similarity in expression differences between reproductive male and female flies and the expression differences between queen and worker pharaoh ants and honey

bees. We think that this is likely to be true. To be more careful and conservative in our interpretation, we have changed the wording to “Our results are consistent with the notion that caste-biased genes are derived from ancient plastically-expressed genes underlying female reproduction” (L: 244-245). We also added “Future studies including more species will be necessary to determine the generality of the patterns (e.g., the precise numbers of shared and lineage-specific genes) we found.” (L: 317-318) to the conclusions to emphasize that more studies are surely needed to assess the generality of our results.

32. Line 227: the authors indicate that no previous comparative study has investigated caste-biased expression in the abdomen. While it’s true that no studies have sequenced multiple species in the same study, there are multiple studies of abdominal queen-worker gene expression differences, many of which have made comparisons across studies posthoc. While these comparisons are less ideal than the current study due to technical differences, as noted, the authors could make an attempt to include at least references to this work in the discussion, and potentially look for hub genes and other overlapping “important” DEGs from this study in those, particularly for additional independent lineages of eusocial insects.

Thank you for pointing this out. We have amended our discussion of the past literature in the introduction to include a citation to two recent such papers (L: 78). In the discussion, we would like to emphasize that no previous study has compared abdominal gene expression differences across multiple origins of eusociality. We agree that post-hoc comparisons are less than ideal and are concerned that adding such analyses adds another potential element of confusion and noise to our analysis.

33. Line 242-244: Perhaps it’s not surprising that honey bees and pharaoh ants show little overlap in nurse-forager DEGs it is known that honey bees show different relationships of some of the canonical signaling pathways involved in division of labor (such as the relationship between JH and Vg). The authors could comment on this

We agree that this is an interesting empirical observation that is consistent with our overall results: highly conserved pathways (e.g. involving JH and Vg signaling) influence division of labor across social insects, but the precise effects (and perhaps downstream molecular mechanisms) differ between lineages. We added the sentence “This could reflect a combination of different roles of nurses and foragers between lineages⁶ as well as differences in the precise molecular relationships between these conserved pathways^{19,53,54}” (L: 272-274), with citations to literature detailing some of these differences.

34. Line 255: missing word after “In contrast”

Added “to”, thank you

35. Line 266: typo, word “of”

Fixed, thank you

36. Line 268: “many traits” do these traits share any particular features in common that make them different from morphological traits? Specific to behavior? Complex traits? Polygenic traits?

In response to a comment from a separate reviewer (comment #69), we have altered this section of the discussion to better incorporate past research on the topic, and have removed this sentence.

37. Line 281: “on top” is semantically confusing, as I believe the authors are referring to TRGs that are downstream in the network

Thank you for this suggestion, we agree this could be confusing. We have changed the text to “layered downstream” (L: 320)

38. Fig. 2A and B: blue worker up-regulated points difficult to see

Thank you for pointing this out. We have made the points bigger to be more visible.

39. Fig 3: “Caste bias is derived from sex bias” – the correlation shown does not convince me of this

We changed wording to “Caste bias is linked to sex bias” to be more conservative

40. Supplement line 44: extra words “were performed”

Fixed, thank you

41. Supplement line 51: missing end parenthesis

Fixed, thank you

42. Supplement line 56: Aculeata should not be italicized

Fixed, thank you

43. Supplement line 84-85: can the authors comment on why honey bee egg samples would cluster with queen abdomen samples? This suggests to me that the abdominal gene expression signal is coming primarily from oocytes in the ovaries.

We agree that this is an interesting difference. We have added a sentence: “It is possible that this difference is a result of a difference of age of the eggs at time of collection: honey bee eggs were 24 hours old and likely still contained maternal RNA, while pharaoh ant eggs were 7 days old.” (L: 90-92) See also our answer to comment #2 for our discussion of active female reproductive physiology.

44. Supplement lines 93 and 96: gene numbers (1039, 1245) are not consistent with those in the main text (lines 149-150, 1006, 1174)

Thank you for pointing this out. The main text numbers were correct and we have fixed the supplemental material.

45. Supplement line 134-135: what is “value” of overall caste bias or overall behavior bias? logFC? Is pvalue uncorrected or corrected for multiple testing?

The P-value is uncorrected (there were only two tests for each species). To clarify the issue of value, we changed the text to: “We identified GO terms associated with overall caste or behavior bias using the Euclidean distance of \log_2 fold-change between queens and workers or nurses and foragers at each stage.” (L: 151-152) to remind readers of the calculation of overall caste bias

46. Fig. S3 legend typo, “indicating” should be “indicate”

Fixed, thank you

47. Fig. S3 please indicate which stage/tissues have significant differences in logFC across phylostrate, because it is not given in main text or figure which (if any) are significant

Thank you for pointing this out. Phylostrata has a significant effect on logFC in each stage/tissue comparison. We have added the text “ \log_2 fold-change varies according to phylostrata for every stage/tissue for each species ($P < 0.001$).” to the figure legend.

48. Fig. S4 clarification needed about “between developmental stages” this is referring only to larval developmental stages, correct?

Added “embryonic and larval” to the caption, thank you for the suggested clarification

Reviewer 2

The authors set out to understand the genetic underpinnings of caste determination in social insects and to determine whether or not species that represent independent origins of eusociality share genetic mechanisms underpinning division of labor. They did this by examining tissue-specific patterns of gene expression in pharaoh ants and honey bees, which represent two different origins of eusociality. They compared the expression profiles of several different developmental timepoints and focal tissues in queens vs workers, nurses vs foragers and males.

49. I am very excited to finally see a study that captures a much more complete range of tissues and timepoints in two different species. It is about time someone did this!

Thank you for your comments, and your enthusiasm for the study! While it is difficult to collect, analyze, and present such a complex dataset, we hope that this study will greatly aid future research in the field.

However, I was surprised that several confounding factors may not have been accounted for in the study design and were not discussed in the manuscript.

The main findings the authors emphasize in the study are that: (1) a large proportion (~30%) of the genes differentially expressed between queens and workers in ants and bees are shared, (2) these genes are hublike with higher levels of connectivity in coexpression networks, (3) these differences are correlated with differences in sexbiased expression, and (4) that these genes are also associated with reproduction in *Drosophila*.

Major comments:

50. To what extent are these differences just driven by the presence of developing eggs inside reproductively-active queens vs non-reproductive workers and males? In other words, how much of this signal is driven by egg expression alone? Developmental genes presumably being expressed in the embryos present inside the queens' ovaries are already known to exhibit all of the above properties and have older evolutionary origins. In fact, the authors find that (at least in the honey bees), many of the queen abdominal samples cluster with egg samples in their co-expression analyses. Because of this, it is unclear to me how much of the findings and results presented in this study are actually driven by caste-based differences between queens and workers and relevant to the evolution of eusociality. This would be more compelling if this

authors could also demonstrate this was true in other tissue types (such as the brain). However, in all other tissues, this pattern does not appear to hold.

Please see our detailed discussion of these issues in regard to questions from Reviewer 1 (in particular answers to comments #2 and #3). As suggested, we demonstrated that our results hold in the case of head and thorax, and added a supplemental figure to depict this (Supplementary Fig. 8), and added text (L: 257-263) to the discussion

51. Instead it seems to be younger, taxon-restricted genes that are differentially expressed, and these differences appear to be unique to each species examined. This, in my opinion, is the most interesting finding in this study. I was bit surprised that this finding was less emphasized than the shared abdominal expression patterns. However, I appreciate the difficulties associated with proving a negative result and the difficulties of making functional comparisons across different species.

Thank you for your comment. We agree, this is an extremely interesting finding. We have tried to be even-handed (e.g. in the title, abstract) in our presentation of these two separate main findings. That said, we ended up spending some some more time discussing the portion of genes with shared expression patterns because our study is the first to find a strong signature (i.e. thousands of genes) consistent with the notion of a reproductive groundplan, which is a leading hypothesis in the literature. Moreover, the lineage-specific finding is somewhat less novel (in the sense that a number of previous studies have emphasized that many differentially-expressed genes, in particular in the worker caste are taxonomically restricted) and perhaps a bit harder to interpret for most readers.

52. It was unclear from the methods whether the authors took any steps to minimize differences associated with the specific life histories of each species. For example, how comparable are behavioral classes across each species? I believe both have an age-based division of labor, but are these dynamics similar in both? For example, are only foragers exposed to light in both species?

There are certainly fundamental differences between the behavioral classes in both species, but also many similarities. The dynamics of age-based division of labor are at least qualitatively similar. To clarify, we added: “Both honey bees⁴² and pharaoh ants³² exhibit age-based worker division of labor, in which younger individuals tend to specialize on nursing and other within-nest activities and older individuals specialize on foraging” to Results (L: 116-118), and “While *M. pharaonis* does exhibit age polyethism with respect to nursing and foraging⁷, the precise dynamics with regard to age are not well studied in comparison to honey bees” to supplemental methods (L: 22-24). We also added the sentence “Future studies including more

species will be necessary to determine the generality of the patterns (e.g., the precise numbers of shared and lineage-specific genes) we found” (L: 317-318).

53. Did the authors make any attempt to control for age-associated differences between nurses and foragers? Many of the behavioral DEGs were associated with development and metabolism, and age can have large effects on these functional groups as well. Similarly, it seems males were only 1 day old; how might this affect the results?

We did not make attempts to disentangle age-associated differences from behavioral differences, in part because by definition age-based division of labor (age polyethism) is closely tied to age. We did previously publish a paper (Mikheyev and Linksvayer 2015 eLife) looking into detail at transcriptomic profiles of workers defined by age and other workers defined by behavior, and as expected there was strong overlap because behavior is strongly age-associated. A large body of literature in honey bees has more thoroughly attempted to disentangle these factors, but we chose to compare nurses and foragers in a relatively “natural” condition, as behavior cannot truly be disentangled from physiological age. With regard to males, we sampled males upon emergence to ensure they had not yet mated and to be consistent across species. Sampling at different ages would be ideal, but was outside the scope of this study.

Minor comments:

54. More details on how nurse/forager classifications were determined would be helpful. Was it just an age-based categorization, or were more detailed behavioral observations made to confirm these behaviors?

We used behavioral observations in *M. pharaonis*. In *A. mellifera*, we paint-marked workers upon emergence and sampled nurses and foragers based on age (nurses less than 7 days old, foragers older than 21 days), given that the dynamics of age polyethism are very well studied in *A. mellifera*. We clarified these issues in (L: 22-26, L: 41-42 (supplemental))

55. More details on the specifics of this analysis of the DEG analysis would be very helpful. It seems the primary data filtration was a minimum CPM < 1 in all samples. Was there a minimum # of samples that had to have CPM>1?

Thank you for pointing this out, we changed the sentence to: “We removed lowly-expressed genes that did not meet one of two criteria: 1) counts per million (CPM) greater than one in at least half the samples, 2) CPM > 1 in all samples of a given tissue/stage/caste combination (to ensure tissue-specific genes were retained)” (L: 343-346)

56. How many of the DEGs had low expression levels?

Added text: “We removed 2350 lowly-expressed genes in ants, leaving 10804 genes for further analysis, and we removed 2036 genes in bees, leaving 11775 genes for further analysis” (L: 346-347)

57. Was there a minimum fold change required to consider something differentially expressed?

No, differential expression was based solely on FDR.

58. Similarly for the coexpression and clustering analyses, more methodological details would have been useful.

We tried to be concise in the main text, but gave more details in supplemental methods (L: 69-107)

59. For the coexpression analyses, what happens when you use the same cutoff for honey bees and ants? Please clarify why two different cutoffs were selected for each species.

We attempted to clarify this in the supplemental methods (L: 97-107). In honey bees, a 90% cutoff contained very few genes, given that two main classes of clusters were found. In ants, a 60% cutoff would be essentially the same as a 90% cutoff given the large peak in the distribution of gene-cluster association after 90% (see Supplementary Figure 14).

60. Please specify the numbers of colonies and individuals for honey bees /ants for each behavioral class; were individuals related or not, especially with respect to queens?

Thank you for the clarification suggestion. We added: “We collected three biological replicates of each specific sample type. Each biological replicate contained a pool of individuals (N = 10 for ants, N = 5 for honey bees) from the same colony, such that each biological replicate corresponds to a colony. The only exception to this was mature honey bee queens, which were sampled from separate, unrelated colonies” (L: 334-338)

Reviewer 3

61. The authors generated queen and worker RNAseq data for a pharaoh ant and a honey bee species, generated from multiple larval instars, pupae, and adult body segments. For adult workers, they also generated separate transcriptomes from adult nursing workers and foraging

workers. The parallel use of common methods and a common sampling scheme represents an important advance over prior meta analyses of castes in ants and bees.

Thank you, we agree that a large strength of the study is the parallel sample collection and analysis, as there are obviously many technological and analytical details that can bias transcriptomic results.

62. Biological replication is somewhat modest, with 3 replicates for each sample type, but the sheer number of sample types considered in the study is an asset, and the statistical analyses and inferences appear sound. Remarkably little RNA-seq data exists that compares queen and worker gene expression in the honey bee *A. mellifera*, and I anticipate this developmental time course data will be of value to future studies, particularly given the importance of *A. mellifera* as a model eusocial insect.

Thank you very much for these positive comments. We agree, there is a paucity of RNA-seq data for these contexts in *A. mellifera*, and this dataset will likely be well-utilized in the future.

63. The manuscript is framed in terms of investigating the extent of molecular convergence exhibited by independently-evolved caste systems in a bee and an ant. As emphasized in the title, the authors observe that many genes fundamental to female reproduction are associated with reproductive division of labor, but they also reconcile this with the existence of much higher numbers of lineage-specific, plastic genes. I find that this treatment of the findings provides useful insight and perspective on hypotheses that have served as a hotbed of recent research by the social insect community. Several suggestions for revisions follow.

Thank you for your interest, positive comments, and careful suggestions.

64. L96: Specify level of biological replication ($n = 3$) in main text – here or in methods.

Added text to the main text methods: “We collected three biological replicates of each specific sample type.” (L: 334-335)

65. The authors used an FDR corrected P-value threshold of < 0.1 to generate very large lists of differentially expressed genes. I’m curious if the trends related to DEG categorization in Figure 1 are robust to more stringent cutoffs. Does the use of $FDR < 0.05$ or a combined FDR + fold-change cutoff alter any of the findings related to detectable orthology or phylostrata in Figure 1?

Thank you for the suggestion. We included a new supplemental figure (Supplementary Fig. 1) which contains results from the analysis using an FDR cut-off of 0.05. It does not qualitatively change the results (beyond the overall numbers of DEGs).

66. Although there are many sample types, the use of 3 biological replicates seems low for a gene coexpression analysis, particularly given that the network module for queen abdominal expression is largely informed by a comparison of 1 sample type (adult queen abdomen) to the others. Can the authors comment on this issue?

We agree this is less than ideal, but the specific algorithm we employed is designed to detect clusters of genes specifically associated with certain samples. Please note that we included all samples (N = 90 for ants, N = 87 for honey bees) in the analysis, so the inferences is also based on the absence of strong co-expression relationships in other sample types. We added “While our level of biological replication (N = 3 for each tissue/caste/stage combination) is low, including all samples in our biclustering analysis allows high resolution of gene co-expression relationships, and biclustering allows for specificity of gene-sample relationships.” to supplemental methods (L: 73-76)

67. Similarly, can the same insights from Figure 2 be provided by analyses of gene expression levels alone (e.g., are DEGs present in both species more likely to have larger $\log_2(Q/W)$ abdominal) values than DEGs in only one species)?

We performed co-expression analysis to identify “hub” genes, which have been shown to be often be functionally associated with traits (in contrast to simply having large lists of differentially expressed genes). This analysis effectively allows us to leverage another dimension (co-expression) to identify which of the ~4000 DEGs are highly important.

68. L190: please provide correlation coefficients in text.

Thank you, we have added coefficients to text

69. Results from Figure 4 showing that degree of gene expression bias is positively correlated with evolutionary rate may warrant citing Hunt et al. 2011 (<https://doi.org/10.1073/pnas.1104825108>) in the discussion given the similarities with these prior findings..

Thank you, we added a citation to the referenced paper (L: 297), as well as a discussion of this and other similar findings in the context of our results (L: 293-307).

70. L265: I don't think theory from either ref 49 or 51 supports an expectation that most gene expression differences among morphs would be found in highly conserved hub genes rather than downstream peripheral genes. Can this part of the discussion be toned down or made more clear?

We have removed this section of the discussion and replaced it with further discussion of how our results fit in with previous work (L: 293-307)

Reviewers' Comments:

Reviewer #1:

None

Reviewer #2:

Remarks to the Author:

The authors have adequately addressed all of my concerns, and I have no additional comments.
Congratulations!

Reviewer #3:

Remarks to the Author:

The authors did a good job addressing my prior comments and I recommend publication.